

# Weak ergodicity breaking in non-Hermitian many-body systems

Qianqian Chen[1], Shuai A. Chen[2] and Zheng Zhu[1,3]⋆

**1** Kavli Institute for Theoretical Sciences, University of Chinese Academy of Sciences,
Beijing 100190, China
**2** The Hong Kong University of Science and Technology, Hong Kong, China
**3** CAS Center for Excellence in Topological Quantum Computation,
University of Chinese Academy of Sciences, Beijing, 100190, China

⋆ zhuzheng@ucas.ac.cn

## Abstract

The recent discovery of persistent revivals in the Rydberg-atom quantum simulator has revealed a weakly ergodicity-breaking mechanism dubbed quantum many-body scars, which are a set of nonthermal states embedded in otherwise thermal spectra. Until now, such a mechanism has been mainly studied in Hermitian systems. Here, we establish the non-Hermitian quantum many-body scars and systematically characterize their nature from dynamic revivals, entanglement entropy, physical observables, and energy level statistics. Notably, we find the non-Hermitian quantum many-body scars exhibit significantly enhanced coherent revival dynamics when approaching the exceptional point. The signatures of non-Hermitian scars switch from the real-energy axis to the imaginary-energy axis after a real-to-complex spectrum transition driven by increasing non-Hermiticity, where an exceptional point and a quantum tricritical point emerge simultaneously. We further examine the stability of non-Hermitian quantum many-body scars against external fields, reveal the non-Hermitian quantum criticality and eventually set up the whole phase diagram. The possible connection to the open quantum many-body systems is also explored. Our findings offer insights for realizing long-lived coherent states in non-Hermitian many-body systems.

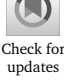

# 1 Introduction

The quantum ergodicity governed by the eigenstate thermalization hypothesis (ETH) depicts most isolated quantum many-body systems locally evolving into an equilibrium statistical ensemble [1–4], and plays a fundamental role in bridging quantum physics and statistical mechanics. Due to the quest for realizing long-lived coherent dynamics, tremendous efforts have been devoted to the ergodicity-breaking mechanisms. It is well-known that, in the presence of an extensive number of conserved quantities, such as the integrable systems [5–8] and many-body localized (MBL) phase [9], the systems fail to thermalize and strongly break ergodicity. In contrast, the quantum many-body scar (QMBS) systems [10–14], which have much fewer numbers of conserved quantities and are free of disorder, exhibit a distinct ergodicity breaking mechanism [15–44]. The QMBS system typically consists of both thermal and nonthermal eigenstates, and is distinguished by specific initial states experiencing periodic revivals, as first observed in an ultra-cold Rydberg atom chain [45].

Until now, the ergodicity-breaking mechanisms are mainly focused on the ideal isolated systems with Hermiticity. Nevertheless, perfect Hermiticity can be broken down [46–54]. The non-Hermiticity could either arise from the non-reciprocal process, such as the cold-atom platforms [55–62] with spontaneous decay or imaginary field, or introduced by the contact with thermal or nonthermal environments, namely the open quantum systems [63–71]. Unlike the Hermitian systems, the study of the ergodicity breaking in many-body non-Hermitian systems remains in the early stage.

The non-Hermiticity hosts many peculiar phenomena [see reviews [53, 71, 72] and reference therein] beyond the Hermitian framework, including complex-valued energy spectra, a biorthonormal basis, the non-Hermitian quantum criticality and the exceptional points (EP) with simultaneous coalescence of eigenvalues and eigenstates. In quantum many-body systems, the interplay between strong interaction and non-Hermiticity may cause unseen thermodynamic phenomena that are far from equilibrium, and thus it is of fundamental importance to ask whether the features decisive for the thermalization/non-thermalization in a Hermitian system still persist when one of the most fundamental premises in quantum mechanics, i.e., Hermiticity, is broken non-perturbatively.

In a different context, the non-Hermiticity is also closely relevant to open quantum systems. In reality, perfect isolated systems with Hermiticity hardly exist due to inevitable contact with the environmental bath. The system coupled to a thermal bath usually relaxes to a Gibbs ensemble with the temperature of bath [64–67], while the system coupled to a nonthermal bath may host distinct thermalization mechanisms [68, 69] due to the arbitrary nonunitary process. Theoretically, the open quantum system is commonly described by the Lindblad master equation [63]. Alternatively, the dynamics of open quantum systems could also be captured by a non-Hermitian Hamiltonian under certain circumstances, for instance, when quantum jumps can be neglected under the postselection [70] or in semi-classical systems [71]. Then the investigation of non-Hermitian many-body Hamiltonian is insightful for open many-body systems.

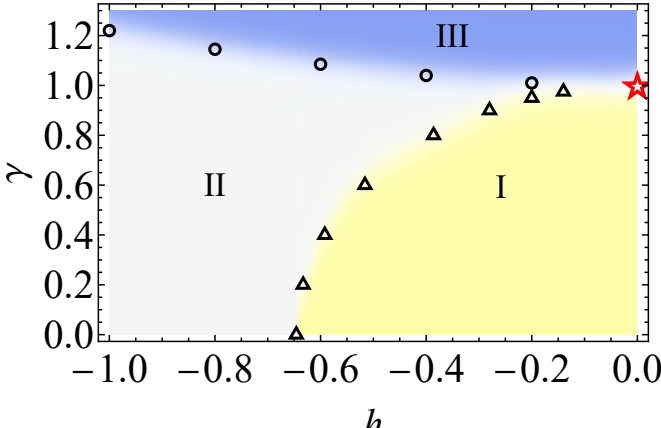

Figure 1: Ground-state phase diagram. We identify three distinct phases (I, II, III) of model Hamiltonian (5) as a function of $\gamma$ and $h$, the phase boundaries of which are determined by the derivatives of the ground-state energy and the fidelity susceptibility. The triangles (circles) are quantum critical points of continuous (first-order) quantum phase transitions driven by the external fields $h$ (strength of non-Hermiticity $\gamma$). The red star denotes the quantum tricritical point, which is an EP and also features a real-to-complex energy spectrum transition. The phase diagram is determined with data steps $\Delta h = 0.005, \Delta \gamma = 0.005$ for $N = 26$ system.

Previous studies have made considerable progress on the ergodicity and strong ergodicity breaking associated with non-Hermitian systems [73–79]. However, there are few studies [41] regarding whether the weak ergodicity breaking exists in the presence of non-Hermiticity both theoretically and experimentally, especially in the non-perturbative regime. Motivated by the above, in this work, we propose a weak ergodicity breaking mechanism in non-Hermitian systems from QMBS in a non-perturbative way, named the non-Hermitian QMBS, and characterize their nature via the periodic many-body revivals in dynamics, the entanglement entropy, the physical observables and the energy-level statistics using both the biorthonormal and self-normal eigenstates [80, 81]. We examine their stability in the presence of external fields, find their critical features when approaching the EP, reveal the non-Hermitian quantum criticality, and finally establish the whole ground-state phase diagram, as illustrated in Fig. 1. Interestingly, we find that the fingerprints of non-Hermitian scarred states switch from the real-energy axis to the imaginary-energy axis after a real-to-complex spectra transition, where an EP and a quantum tricritical point emerge simultaneously. In particular, we find that the non-Hermitian QMBS exhibit longer revival periods and become fragile as the non-Hermiticity strength increases in the real-spectrum region. The insights into the scarred dynamics in an open system are also discussed. Our construction of the non-Hermitian QMBS might be probed via the newly developed measurement of the non-Hermiticity or future advanced techniques in ultracold-atom platforms [45, 55–61].

## 2 Model setup and Methods

Before establishing the non-Hermitian quantum many-body scar, we first introduce the Hermitian counterparts realized in a Rydberg-atom quantum simulator [45, 82] with the Hamiltonian

$$H_{\text{Ryd}} = \sum_{i=1}^{N} \left( \frac{\Omega}{2} \sigma_i^x + \Delta n_i \right) + \sum_{i<j} V_{i,j} n_i n_j \,. \tag{1}$$

Here, $\sigma_i^x = |g_i\rangle\langle r_i| + |r_i\rangle\langle g_i|$ couples an atom between the ground state $|g_i\rangle$ and the Rydberg excited state $|r_i\rangle$ at position $i$, which is realized by a two-photon transition and driven at Rabi frequency $\Omega$. $\Delta$ denotes the strength of the laser detuning and $n_i = |r_i\rangle\langle r_i|$. The potential $V_{i,j} \propto 1/R_{i,j}^6$ characterizes the van der Waals interaction between atoms in Rydberg states at a distance $R_{i,j}$. In the limit of strong nearest-neighbor interactions $V_{i,i+1} \gg \Omega$, the system can be effectively described by

$$H_{\text{PXP}} = \sum_i^N P_{i-1}\sigma_i^x P_{i+1}\,, \tag{2}$$

without simultaneous Rydberg excitations of nearest neighbors [10,11], where $P_i = (1-\sigma_i^z)/2$ is a projection operator with $\sigma_i^z = |r_i\rangle\langle r_i| - |g_i\rangle\langle g_i|$. Here we introduce the non-Hermiticity by generalizing the symmetric coupling between $|r_i\rangle$ and $|g_i\rangle$ to be non-reciprocal, i.e.,

$$|g_i\rangle\langle r_i| + |r_i\rangle\langle g_i| \rightarrow (1-\gamma)|g_i\rangle\langle r_i| + (1+\gamma)|r_i\rangle\langle g_i|\,, \tag{3}$$

so as to yield a non-Hermitian many-body Hamiltonian,

$$H_{\text{nH-PXP}} = \sum_j^N P_{j-1}\left(\sigma_j^x + i\gamma\sigma_j^y\right)P_{j+1}\,, \tag{4}$$

where the parameter $\gamma \in \mathbb{R}$ denotes the strength of non-Hermiticity. Incidentally, the non-Hermitian term in Eq. (4) could also be regarded as the application of an imaginary magnetic field [62]. In the following, besides establishing and characterizing the non-Hermitian QMBS states, we will also examine their stability against an external magnetic field $h \in \mathbb{R}$. The whole Hamiltonian can be written as

$$H = H_{\text{nH-PXP}} + \sum_j^N h\sigma_j^z\,. \tag{5}$$

The non-Hermitian Hamiltonian $H$ has the right eigenstates $\{|\psi_{\text{R},n}\rangle\}$ and left eigenstates $\{|\psi_{\text{L},n}\rangle\}$ [71,80]

$$H\left|\psi_{\text{R},n}\right\rangle = E_n\left|\psi_{\text{R},n}\right\rangle \quad \text{and} \quad H^\dagger\left|\psi_{\text{L},n}\right\rangle = E_n^*\left|\psi_{\text{L},n}\right\rangle\,. \tag{6}$$

There is often only an orthogonal relationship between the left and right eigenvectors for eigenstates; this relationship is known as biorthogonality. Together with the normalization condition, we have the biorthonormal relation

$$\langle\psi_{\text{L},n}|\psi_{\text{R},m}\rangle = \delta_{nm}\,.$$

Due to non-Hermiticity, in general, the right (left) eigenstates may not keep orthogonality $\langle\psi_{\text{R},n}|\psi_{\text{R},m}\rangle \neq 0$ ($\langle\psi_{\text{L},n}|\psi_{\text{L},m}\rangle \neq 0$) for $m \neq n$. In terms of the biorthogonal basis, we can have a spectral decomposition

$$H = \sum_n E_n|\psi_{\text{R},n}\rangle\langle\psi_{\text{L},n}|\,.$$

To examine the whole spectrum of the model Hamiltonian (5) as a function of $\gamma, h$, we use the exact diagonalization (ED) approach. We utilize both the translational symmetry and the spatial inversion symmetry under the periodic boundary condition (PBC), which enables us to fully diagonalize the Hamiltonian up to $N = 32$ sites. The quantum number of the momentum is labelled as $k$ ($k = 2\pi m/N$ with $m = -N/2, \cdots, N/2$), while the inversion symmetric or anti-symmetric sector is denoted as $I$ ($I = +, -$). The quantum many-body scarred states are located in the $(k, I) = (0, +)$ and $(k, I) = (\pi, -)$ sectors, both of which give similar results, we therefore mainly focus on the $(k, I) = (0, +)$ sector with Hilbert space dimension up to $D = 77,436$ for $N = 32$ system.

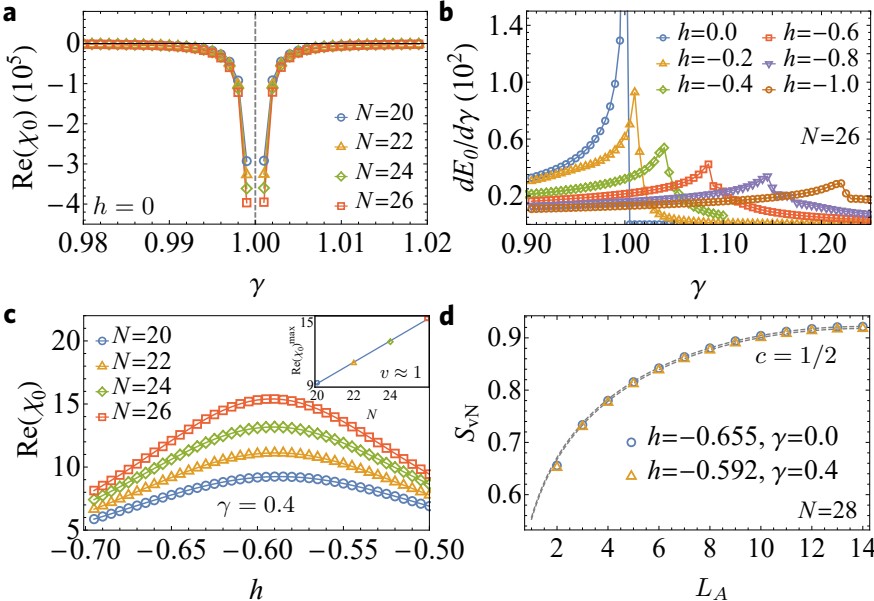

Figure 2: EP and quantum critical points. **a** The real part of the generalized fidelity susceptibility $\mathrm{Re}(\chi_0)$ with respect to the strength of non-Hermiticity $\gamma$. We remark that the real parts of the eigenenergies are all zero for $\gamma > 1$ and $h = 0$. Thus, at such parameter region, to depict the behavior of the states with imaginary eigenenergies, here we plot the fidelity susceptibility of the states with the lowest imaginary parts of the eigenenergies. The divergence of $\mathrm{Re}(\chi_0)$ towards negative infinity when approaching $\gamma_c = 1$ signifies an EP. **b** The first order derivative of the ground-state energy with different fixed magnetic fields $h$ and the singularity characterize the quantum phase transitions driven by tuning the strength of non-Hermiticity $\gamma$. **c** The real part of the generalized fidelity susceptibility $\mathrm{Re}(\chi_0)$ as a function of the external fields $h$. The inset shows the finite-size scaling of the maxima of $\mathrm{Re}(\chi_0)$, where linear fit suggests the critical exponent $\nu \approx 1$. **d** The entanglement entropy of the ground states $S_{\mathrm{vN}}$ as a function of the subsystem length $L_A$ for $N = 28$ system. At the critical points, the numerical data can be fitted through $S_{\mathrm{vN}} \sim c/3 \ln(\sin(\pi L_A/N)) + \mathrm{const.}$ (dashed gray lines) with the central charge $c = 1/2$.

## 3 Results

### 3.1 Non-Hermitian quantum criticality

Before examining the excited-state properties, we begin with establishing the ground-state phase diagram in the $\gamma$-$h$ plane (see Fig. 1) using the biorthonormal eigenstates. Here, for the consistency of the definition, we define the ground state as the state with the smallest real eigenenergy like in Hermitian systems, and our main focus phase (I) in Fig. 1 has fully real eigenvalues.

As shown in Fig. 1, the black triangles and the circles denote the boundaries of phases (I, II, III). A quantum tricritical point (red star) emerges at $(\gamma, h) = (1, 0)$, which is also a real-to-complex spectrum transition point and an EP simultaneously. To examine the phase boundaries, we compute both the derivatives of the ground-state energy and the fidelity susceptibility. We adopt the generalized fidelity $\mathcal{F}$ for non-Hermitian systems

$$\mathcal{F}(\lambda, \delta\lambda) = \langle \psi_L(\lambda) | \psi_R(\lambda + \delta\lambda) \rangle \langle \psi_L(\lambda + \delta\lambda) | \psi_R(\lambda) \rangle. \tag{7}$$

Similar to the Hermitian case, the relation of the fidelity $\mathcal{F}$ and its corresponding fidelity

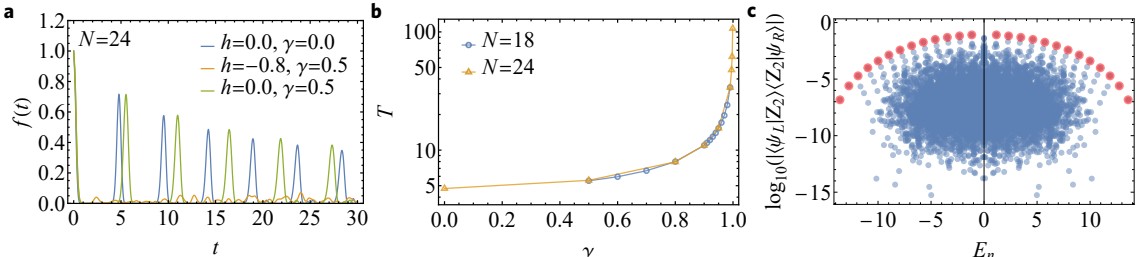

Figure 3: The quantum fidelity in dynamics. **a** The fidelity dynamics of initial $|Z_2\rangle$ state $f(t) = |\langle Z_2|Z_2(t)\rangle|^2$ exhibits coherent periodic revivals with period $T$ for parameters of non-Hermitian QMBS (green) but collapses abruptly for a larger $h$ (yellow). The Hermitian case (blue) is depicted for comparison. **b** The revival period $T$ as a function of the non-Hermiticity strength $\gamma$ at $h = 0$, which exhibits divergent behavior when approaching the EP $(\gamma, h) = (1, 0)$. **c** The overlap between eigenstates and the $|Z_2\rangle$ state with respect to eigenenergies. Red dots denote non-Hermitian quantum many-body scarred states with approximate equal energy spacing $\Delta E = 2\pi/T$. Data shown are for $N = 26$ and $\gamma = 0.5, h = 0$ in the zero and $\pi$-momentum sector.

susceptibility $\chi$ is $\mathcal{F} = 1 - \chi\delta\lambda^2 + O(\delta\lambda^3)$. Therefore, the fidelity susceptibility $\chi$ can be approximated as

$$\chi(\lambda) \approx (1 - \mathcal{F})/\delta\lambda^2. \tag{8}$$

Physically, the divergence of $\chi$ towards negative infinity implies EPs [83–85].

We first probe the quantum phase transitions when tuning the non-Hermiticity strength $\gamma$. At $h = 0$, as shown in Fig. 2a, the real parts of the ground-state fidelity susceptibility $\mathrm{Re}(\chi_0) \to -\infty$ with increasing system length when approaching $\gamma_c = 1$, which demonstrates an EP. Such an EP is notable because not only do the eigenvalues coalesce there, but also the corresponding eigenstates become entirely parallel, which is impossible in a standard Hermitian system. We also find that the first-order derivative curve $dE_0/d\gamma$ displays a singularity at $\gamma_c$ (see Fig. 2b). These observations of EP are consistent with the evolution of the energy spectra, where the whole real spectra at $\gamma < 1$ develop into complex conjugate pairs at $\gamma > 1$, separated by a transition at $\gamma_c = 1$. The quantum phase transitions at other fixed fields $h$ can also be identified similarly [see Fig. 2b], as denoted by the circles in Fig. 1.

By contrast, we find the Hamiltonian exhibits a continuous phase transition when tuning the magnetic field $h$ at $\gamma < 1$. Figure 2c shows $\mathrm{Re}(\chi_0)$ as a function of $h$ for different system lengths $N$, where the smooth curve as well as the increasing maximal values of $\mathrm{Re}(\chi_0)$ with $N$ indicate the continuity of such a phase transition, which can also be confirmed by the absence of singularities in the first-order derivative curve $dE_0/dh$. We further analyze the critical exponent $\nu$ and the central charge $c$ in a manner similar to the Hermitian case. The critical exponent $\nu$ from the finite-size scaling theory can be directly extracted from the fidelity susceptibility via $\mathrm{Re}(\chi_0)^{\mathrm{max}} = N^{2/\nu - 1}$ [81, 86], as shown in the inset of Fig. 2c, where the linear fitting demonstrates $\nu = 1$. The central charge of the conformal field theory at the critical point can be obtained from the scaling of generic bipartite von Neumann entanglement entropies defined for non-Hermitian systems (see the definition Eq. (11) below) through

$$S_{\mathrm{vN}} \sim c/3 \ln(\sin(\pi L_A/N)) + \mathrm{const.}, \tag{9}$$

where $L_A$ is subsystem lengths. Such a central charge determines the universality class of the quantum criticality. We find that for our non-Hermitian model, the logarithmic fitting in Fig. 2d indicates a finite central charge $c = 1/2$. Here we notice that the critical exponent $\nu$ and the central charge $c$ are the same as the phase transition reported in the Hermitian limit

(i.e., $\gamma = 0$) [87–90], indicating the same universality class might be generalized to the non-Hermitian case. When $\gamma = 0$ and $h \to \infty$, the ground states are two-fold degenerate due to the projection constraint, both breaking the $\mathbb{Z}_2$ symmetry and violating thermalization. In particular, we find a quantum tricritical point when the above two phase transitions meet at the EP $(\gamma, h) = (1, 0)$.

## 3.2 The periodic revivals and overlaps of specific states

We have established the ground-state phase diagram above. Below we further examine the properties of the excited states and the non-Hermitian QMBS states.

The existence of many-body scarred states can be inferred by the periodic revival of the quantum fidelity for the experimentally realizable antiferromagnetic state $|Z_2\rangle \equiv |r_1 g_2 r_3 g_4 \dots\rangle$ (c.f. Fig. 3a), manifesting a long-time memory of the initial state in the non-Hermitian many-body systems. The revival period $T$ corresponds to a characteristic energy interval $\Delta E$ of the scarred states through the relation $T = 2\pi/\Delta E$. Strikingly, we find such revival period $T$ tends to diverge when approaching the EP $(\gamma, h) = (1, 0)$, as depicted in Fig. 3b. This observation suggests the enhanced coherence time in revival dynamics with increasing the non-Hermiticity strength towards an EP.

Moreover, the eigenstates that have a large overlap with the initial product state $|Z_2\rangle$ are embedded in the energy spectrum with approximate same energy spacing $\Delta E$, as depicted by red dots in Fig. 3c. Such overlap is defined by $|\langle \psi_L | Z_2 \rangle \langle Z_2 | \psi_R \rangle|$ with the biorthonormal basis. The number of such states is equal to the system size $N$. Since the features of these eigenstates in the non-Hermitian many-body system are similar to the Hermitian QMBS, we dub them non-Hermitian QMBS. Besides exhibiting a longer revival period $T$ compared to the Hermitian PXP model at $(\gamma, h) = (0, 0)$, the non-Hermitian QMBS also become more fragile against the external field $h$ (see the yellow line in Fig. 3a) with increasing non-Hermitian strength $\gamma$.

## 3.3 Entanglement entropy and physical observables

Below we will show that the non-Hermitian QMBS can also be characterized by the abnormally low entanglement entropy and the eigenstate expectations of physical observables that deviate from those in the bulk spectrum.

In Hermitian many-body systems, entanglement entropy is a complementary way to examine thermalization. Here, we generalize the von Neumann entanglement entropy to non-Hermitian many-body systems. We first consider the non-Hermitian density matrix of the $n$th state $\rho_n$ in terms of the biorthonormal basis

$$\rho_n = |\psi_{R,n}\rangle \langle \psi_{L,n}| , \tag{10}$$

and study a generic entanglement entropy

$$S_{\mathrm{vN}} = -\mathrm{Tr}_A \left( \rho_{A,n} \ln |\rho_{A,n}| \right) , \tag{11}$$

in the non-Hermitian many-body system [91–93]. Here, $\rho_{A,n}$ is the reduced density matrix for subsystem $A$ after tracing out the rest of the system. The generic entanglement entropy Eq. (11) can be reduced to the traditional entanglement entropy in the Hermitian limit and can capture the necessary characteristics in non-Hermitian critical systems [91–93]. Figure 4a shows one typical example of the generic $S_{\mathrm{vN}}$ for the non-Hermitian QMBS at $\gamma = 0.1$ and $h = 0$. In contrast to the highly entangled state, there are a set of eigenstates that exhibit abnormally low entanglement with approximately equal energy difference, as marked by the red dots. Here, we point out that these non-Hermitian scarred states are all located at the

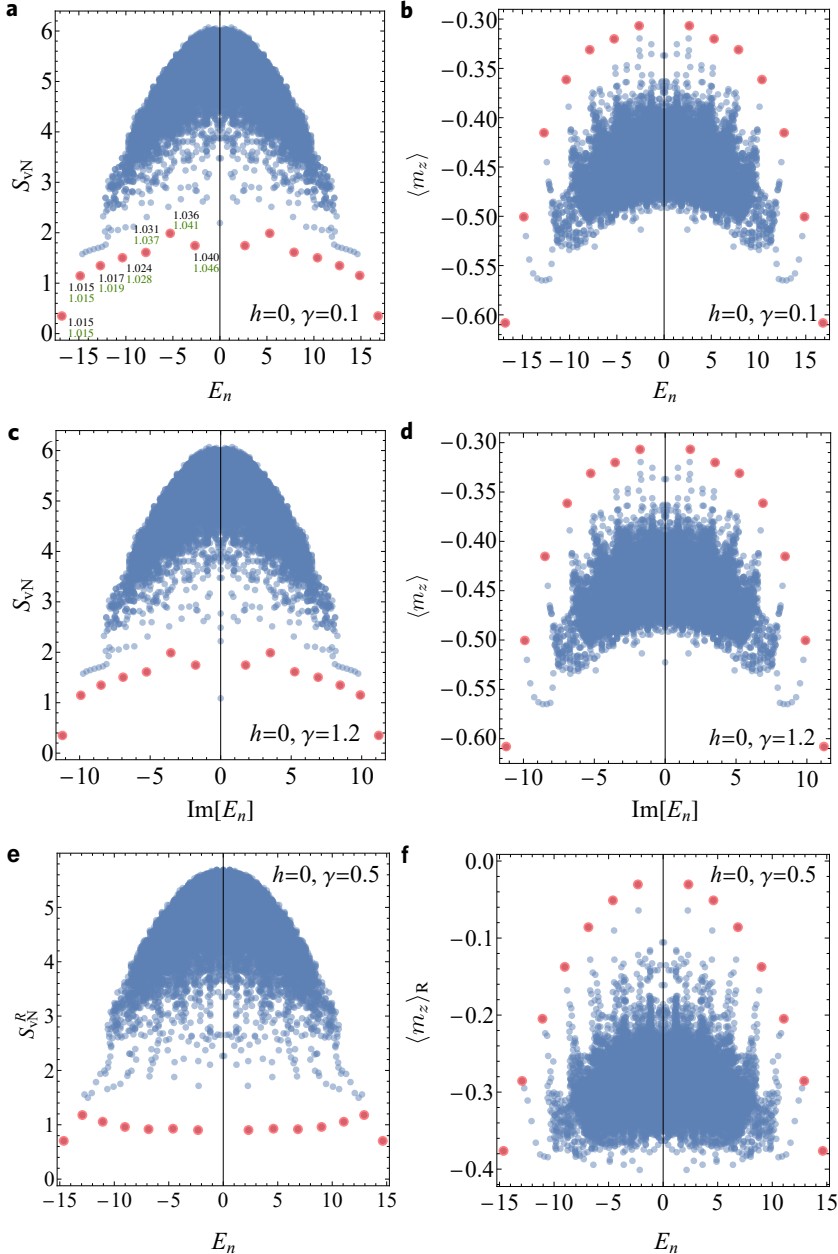

Figure 4: Entanglement entropies and physical observables. Panels (**a, c, e**) show the generalized bipartite von Neumann entanglement entropies $S_{vN}$ (**a, c**) and $S_{vN}^R$ (**e**) with respect to eigenenergies for all eigenstates. The non-Hermitian quantum many-body scarred states exhibit lower entanglement entropy and are highlighted by red dots. The values of overlaps between scar states at $\gamma = 0$ and the corresponding left (right) scar states at $\gamma = 0.1$ are written in black (green) color in **a** alongside the associated states. Panels (**b, d, f**) show the generalized expectations of local observables $\langle m_z \rangle$ (**b, d**) and $\langle m_z \rangle_R$ (**f**), the scarred states (red dots) are clearly distinct from other states and are exactly the ones identified from the entanglement entropy. All the calculations presented here are for the systems with length $N = 28$ in the momentum sector $k = 0$ at $h = 0$.

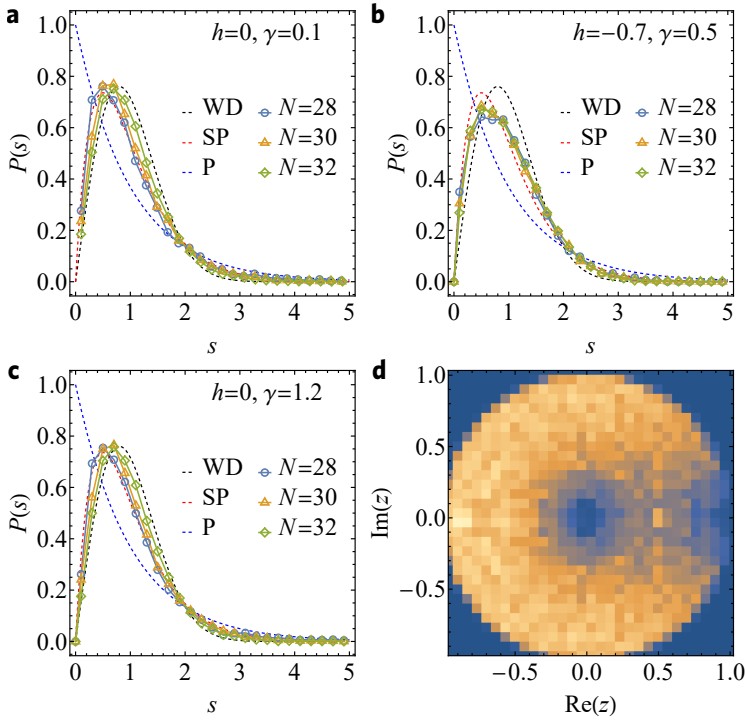

Figure 5: Statistics of energy level spacing with $h = 0$, $\gamma = 0.1$ (**a**), $h = -0.7$, $\gamma = 0.5$ (**b**), $h = 0$, $\gamma = 1.2$ (**c**). As a comparison, Wigner-Dyson (WD) statistics of the Gaussian orthogonal ensemble (GOE) $\pi s/2 e^{-\pi s^2/4}$ (dashed black), semi-Poisson (SP) statistics $4s^{-2s}$ (dashed red) and Poisson (P) statistics $e^{-s}$ (dashed blue) are plotted. Here, we consider the zero-momentum inversion-symmetric $(k, I) = (0, +)$ sector under periodic boundary condition (PBC), for which we eliminate 20% of the eigenenergies found at the spectrum's edges and perform the spectrum unfolding. **d** Density plot of the complex ratio $z$ at $h = -0.3, \gamma = 1.2$ in the complex plane for the systems with length $N = 32$. Darker colors imply a lower density of the ratio $z$. The suppressed ratio density around the origin and for small angles of $z$ suggests significant level repulsion.

$k = 0, \pi$ momentum sectors, which also resemble the Hermitian counterparts. Notably, we find large overlaps between the scarred states at $\gamma = 0$ and the corresponding left (right) scarred states at $\gamma = 0.1$. The values of such overlaps are indicated in black (green) color in Fig. 4a. When approaching the EP $(\gamma, h) = (1, 0)$ along $h = 0$, such typical entanglement entropy structure of non-Hermitian QMBS is still persistent. We also consider a definition of density matrix more similar to that in Hermitian systems, i.e., the Hermitian self-normal density matrix $\rho_n^R = |\psi_{R,n}\rangle\langle\psi_{R,n}|$ [81, 92] with the self-normal assumption $\langle\psi_{R,n}|\psi_{R,n}\rangle = 1$, where non-Hermitian systems serve as effective models of dissipative dynamics without quantum jumps. We show the self-normal entanglement entropy

$$S_{vN}^R = -\text{Tr}_A\left(\rho_{A,n}^R \ln\left|\rho_{A,n}^R\right|\right),\tag{12}$$

at $(h, \gamma) = (0, 0.5)$ in Fig. 4e, where the typical low entanglement outliers with nearly equal energy difference (marked by red dots) are still existent. These results illustrate that the features of QMBS states are relatively robust over the non-Hermitian parameter space with both biorthonormal eigenstates and self-normal eigenstates.

Physical observables can also identify the violation of the thermalization for these non-Hermitian scarred states. We examine the expectation value of the magnetization

$m_z \equiv \sum_i \sigma_i^z / N$ of all eigenstates. Here, both the expectation value with the biorthonormal eigenstates $\langle m_z \rangle \equiv \langle \psi_L | m_z | \psi_R \rangle$ and self-normal eigenstates $\langle m_z \rangle_R \equiv \langle \psi_R | m_z | \psi_R \rangle$ are considered. Due to the translation symmetry, we have $\langle m_z \rangle = \langle \psi_L | \sigma_1^z | \psi_R \rangle$ and $\langle m_z \rangle_R = \langle \psi_R | \sigma_1^z | \psi_R \rangle$. As shown in Fig. 4b, f, we find that a series of states (denoted by red dots) with maximal $\langle m_z \rangle$ and $\langle m_z \rangle_R$ have approximately equal energy-level spacing, and are distinct from other states clearly, similar to the violation of ETH in Hermitian systems. In particular, these special states are exactly the non-Hermitian QMBS states characterized by the low entanglement entropy in Fig. 4a,e. Thus, these eigenstate characters decisive for the weak ergodicity breaking in a Hermitian system still exist in non-Hermitian many-body physics.

Interestingly, in the case of complex energy spectra at $\gamma > 1, h = 0$, we also find a set of eigenstates (marked as red dots in Fig. 4c,d) with low entanglement entropy and maximal expectation values of $\langle m_z \rangle$ but with respect to the imaginary part of eigenenergy. These states are analogous to the non-Hermitian scarred states at $\gamma < 1$.

## 3.4 Energy level statistics

Besides the above-mentioned characteristics of the non-Hermitian QMBS, in this section, we further reveal the chaotic spectrum background they are embedded in by the eigenenergy level-spacing distributions and ratios.

Figures 5a-c show the nearest-level-spacing distribution $P(s)$ of the statistics parameter $s_n = |E_{n+1} - E_n|$ for real or imaginary spectrum, where the $E_n$ are increasingly ordered according to the real or imaginary part, respectively. At $h = 0, \gamma = 0.1$ with non-Hermitian QMBS, although the scarred states have low entanglement entropy (see Fig. 4a), the bulk of the unfolded level statistics displays a Wigner-Dyson (WD) distribution [94, 95], as shown in Fig. 5a, indicating a prominent feature of the quantum chaos, which is fundamentally different from previously known examples of non-ergodicity in non-Hermitian systems [73]. Here, we remark that a chaotic non-Hermitian system is expected to follow Ginibre statistics [75]. However, when the complex spectra become totally real, one may still expect the WD distribution for a chaotic system [75].

At $h = -0.7, \gamma = 0.5$, we find the level-spacing statistics resemble semi-Poisson (SP) statistics near the criticality [c.f. Fig. 5b], at least for the largest system we have reached. As non-universal statistics, SP distribution displays the intermediate statistics between Poisson and WD, and it is typical of the pseudointegrable systems [96]. Here the slight deviation from SP might be induced by the finite size effect. Remarkably, when the whole spectra become imaginary at $\gamma > 1, h = 0$ in phase (III), Fig. 5c shows that the level statistics tend to approach the WD distribution at $h = 0, \gamma = 1.2$, exhibiting the same feature of non-Hermitian QMBS.

The energy spectra become complex at finite external field $h$ in phase (III). To avoid the ambiguity of unfolding complex spectrum, we instead consider the complex level-spacing ratios [97]

$$z_n = \frac{E_n^{NN} - E_n}{E_n^{NNN} - E_n}. \tag{13}$$

Here, $E_n$ is referred to as the $n$th real or complex eigenenergy, of which the nearest neighbor and next-to-nearest neighbor in the complex plane are $E_n^{NN}$ and $E_n^{NNN}$ respectively. Moreover, when the distribution of $z$ is anisotropic, we also consider the radial and angular marginal distribution with the relation $z \equiv r e^{i\theta}$. We calculate the radial and angular marginal distributions of the complex ratio $z$ based on

$$P(r) = \int d\theta\, r P(r, \theta), \quad P(\theta) = \int dr\, r P(r, \theta), \tag{14}$$

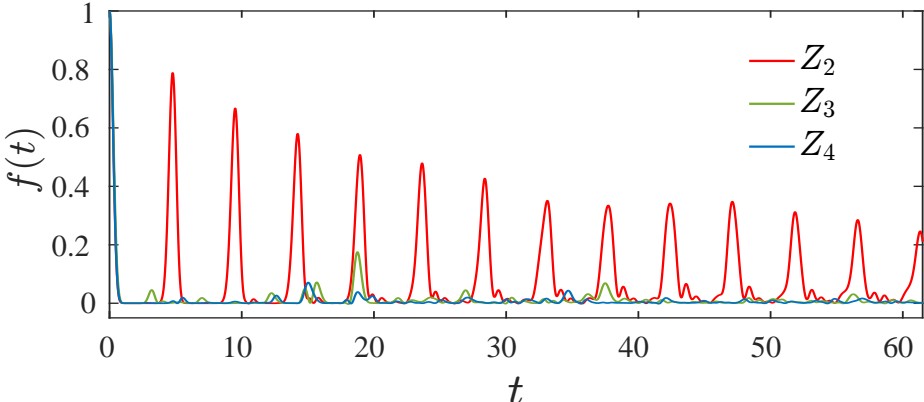

Figure 6: Quantum fidelity by the Lindblad master equation. Evolution of the quantum fidelity $f(t)$ exhibits pronounced revivals for initial product state $|Z_2\rangle$, similar to the periodic revivals obtained by the non-Hermitian model in Eq. (5). By contrast, other initial states like $|Z_3\rangle$ and $|Z_4\rangle$ show a complete absence of quantum revivals. Here, parameters are $\gamma = 0.5$ and $h = 0$ for the $N = 14$ system with periodic boundary conditions.

Table 1: Averaged $\langle r \rangle$ and $\langle \cos\theta \rangle$ for different system sizes $N$. Typical values of Ginibre and Poisson distribution are also given. The averaged values $\langle \cos\theta \rangle$ converge to Ginibre distribution with an increasing $N$.

|  | $N = 32$ | $N = 30$ | $N = 28$ | Ginibre | Poisson |
|---|---|---|---|---|---|
| $\langle r \rangle$ | 0.737 | 0.753 | 0.737 | 0.74 | $\frac{2}{3} \approx 0.667$ |
| $\langle \cos\theta \rangle$ | $-0.181$ | $-0.150$ | $-0.119$ | $-0.24$ | 0 |

to get the means

$$\langle r \rangle = \int dr\, P(r)\, r, \quad \langle \cos\theta \rangle = \int d\theta\, P(\theta) \cos\theta. \tag{15}$$

Here, statistics of the complex ratio $z$ are taken from eigenenergies lying within 80% of the real and imaginary parts from the middle of the spectrum in the zero-momentum inversion-symmetric $(0, +)$ sector. As shown in Fig. 5d for $(h, \gamma) = (-0.3, 1.2)$, we find strongly suppressed ratio density around the origin and for small angles of $z$, demonstrating significant level repulsion. We further compare the averaged $\langle r \rangle$ and $\langle \cos\theta \rangle$ with Ginibre and Poisson distributions. As shown in Table 1, there is an apparent convergence of our results to the average values of Ginibre distribution with the increase of $N$. Our results of the complex level spacing ratios numerically demonstrate that the energy spectrum at $(h, \gamma) = (-0.3, 1.2)$ shows Ginibre distribution, a key feature of quantum chaos in systems with complex spectra.

## 3.5 Connection to open quantum many-body systems

Now we discuss the potential connection of the non-Hermitian QMBS to the open quantum many-body systems. We will show that, the weak ergodicity breaking in our non-Hermitian model is insightful for slow relaxation dynamics of certain initial states in open quantum many-body systems [98] under certain circumstance.

Generally, an open quantum many-body system can be described by the Lindblad master equation [63, 70]

$$\dot{\rho} = \mathcal{L}\rho = -i\left(H_{\text{nH}}\rho - \rho H_{\text{nH}}^\dagger\right) + \mathcal{D}(\rho), \tag{16}$$

where $\mathcal{L}$ is the Liouvillian superoperator, $\mathcal{D}(\rho) = \sum_j L_j \rho L_j^\dagger$ denotes quantum jumps, and $H_{\text{nH}}$ corresponds to an effective non-Hermitian Hamiltonian. The Lindblad operators $L_i$ are related to the non-Hermitian terms in $H_{\text{nH}}$ via (see details in Appendix)

$$-\frac{i}{2} L_j^\dagger L_j = i\gamma \left( g - P_{j-1} \sigma_j^y P_{j+1} \right), \tag{17}$$

where $g$ is a purely imaginary constant to ensure the semi-positive definiteness. We compute the evolution of quantum fidelity for certain initial states by the Lindblad master equation and compare their relaxation dynamics with those obtained from our non-Hermitian model in Eq. (5). Remarkably, as shown in Fig. 6 for $\gamma = 0.5$, the experimentally realizable $|Z_2\rangle$ initial product state [45, 82] exhibits long-time periodic revivals when a constant shift $g$ is large, similar to the behavior of quantum many-body scars shown in Fig. 3a. By contrast, for other initial states like $|Z_3\rangle$ and $|Z_4\rangle$, we do not find any pronounced revivals in the fidelity. To understand such observations, we provide an alternative interpretation based on a perturbative analysis in the large $g$ limit, where the quantum jumps and non-Hermiticity get suppressed (see Appendix for details), leading to an effective description of open quantum systems in the weak non-Hermiticity limit of Eq. (5). We remark that while it may be challenging to experimentally realize the Lindblad operators we use here in our numerical calculation, other Lindblad operators satisfying Eq. (17) might be constructed to be experimentally realizable.

## 4 Summary and Discussion

In this work, we study the exemplary mechanism of the weak ergodicity breaking in non-Hermitian many-body systems, named as non-Hermitian QMBS, in a non-perturbative manner. Using both the biorthonormal and self-normal eigenstates, we systematically characterize the non-Hermitian QMBS from the perspectives of the quantum revivals in dynamics, the eigenstate entanglement entropy in quantum information, the physical observables measurable in quantum simulation, and quantum chaos in statistical physics. Moreover, we also illustrate the robustness of non-Hermitian QMBS both against external field and the influence of EP, reveal the non-Hermitian quantum criticality and establish the whole ground-state phase diagram. Notably, in contrast to the Hermitian QMBS, we find the enhanced coherent revival dynamics of non-Hermitian QMBS near the EP. Finally, the instructive insights on scarred quantum dynamics in open quantum many-body systems are provided and the connection is analyzed in the perturbative limit.

For the Hermitian QMBS realized in the Rydberg-atom quantum simulator [45], the symmetric coupling between the ground state and the Rydberg state is induced by a two-photon process via an intermediate level. Recently, much effort has been devoted to realizing the non-Hermiticity in cold-atom platforms [55–61], notably the Rydberg atoms [58, 59]. By tuning the coupling non-symmetrically [61], or by applying the imaginary magnetic field [62], our results are likely to be verified in recent experimental platforms or in future developed platforms. The non-Hermitian QMBS proposed in this work may also stimulate more experimental activities to realize long-lived coherent states storing the initial quantum information in open quantum many-body systems.

Our work might serve as a starting point to investigate the weak ergodicity breaking mechanism in the absence of Hermiticity. It would be interesting to study the quantum many-body scars in other non-Hermitian many-body systems or models, which may have distinct mechanisms from the non-Hermitian PXP model. Further investigations on the connections of non-Hermitian QMBS to open systems with experimentally realizable Lindblad operators and beyond the perturbative limit is also fundamentally important to deepen our understanding. Our

findings might also motivate future theoretical studies on the eigenmodes of the Liouvillian with purely imaginary eigenvalues [98] from the perspective of the non-Hermitian QMBS. In addition, our work may stimulate future studies on the interplay among thermalization, the non-Hermiticity, and the strong correlations in many-body systems.

## Acknowledgements

We thank Wen-Jie Wei for the helpful discussion.

**Funding information** This work was supported by the National Natural Science Foundation of China (Grant No. 12074375), the Innovation Program for Quantum Science and Technology (Grant No. 2-6), the Fundamental Research Funds for the Central Universities and the Strategic Priority Research Program of CAS (Grant No.XDB33000000).

**Author contributions** Q.C. and S.A.C. contributed equally to this work.

## A Correspondence to a master equation of Lindblad form.

In this appendix, we give details for the correspondence between the non-Hermitian Hamiltonian and a master equation of Lindblad form. We present the explanatory details on the correspondence between the non-Hermitian model in Eq. (5) and an open system whose dynamics can be described by a master equation of the Lindblad form.

A master equation of the Lindblad form describes quantum dynamics of an open system that is bathed in an environment under Markov approximation. In general, we can write the master equation as (with $\hbar = 1$)

$$\dot{\rho} = -i\left(H_{\text{nH}}\rho - \rho H_{\text{nH}}^{\dagger}\right) + \mathcal{D}(\rho),\tag{A1}$$

where $H_{\text{nH}}$ is a non-Hermitian Hamiltonian and is composed of two parts: the Hermitian $H_0$ and the non-Hermitian $-\frac{i}{2}L_j^{\dagger}L_j$,

$$H_{\text{nH}} = H_0 - \sum_j \frac{i}{2}L_j^{\dagger}L_j,\tag{A2}$$

with $L_j$ being the Lindblad operators. The last term $\mathcal{D}(\rho) = \sum_j L_j\rho L_j^{\dagger}$ in Eq. (A1) are quantum jump terms. When the quantum jump term can be neglected, the $H_{\text{nH}}$ will play a role in dominating the dynamical processes.

The main step towards the correspondence is to construct the Lindblad operator with respect to the non-Hermitian Hamiltonian in Eq. (5) in the main text. For this purpose, we can introduce a constant shift term $g$ to our model in Eq. (5) in the main text to ensure $g - P_{j-1}\sigma_j^y P_{j+1}$ semi-definite, and then the Lindblad operators $L_j$ can be determined by the following equation,

$$-i\gamma\left(g - P_{j-1}\sigma_j^y P_{j+1}\right) = -\frac{i}{2}L_j^{\dagger}L_j.\tag{A3}$$

We remark that the constant shift term $g$ imposes no influence on the eigenstates. When $g > 1$, we give one of the examples of the Lindblad operators $L_j$ satisfying Eq. (A3) perturbatively. For the leading orders, we have

$$L_j = \sqrt{2g\gamma}\left[1 - \frac{1}{2g}\left(1 + \frac{1}{8g^2}\right)P_{j-1}\sigma_j^y P_{j+1} - \frac{1}{8g^2}P_{j-1}P_{j+1}\right] + \mathcal{O}\left(\frac{\sqrt{g}}{g^3}\right).\tag{A4}$$

As a directed consequence, the quantum jump terms $\mathcal{D}(\rho)$ further make extra contributions to non-Hermitian terms by observing

$$
\begin{aligned}
\mathcal{D}(\rho) = \gamma \sum_j \Bigg[ 2g\rho - \Bigg\{ (1 + \frac{1}{8g^2}) P_{j-1}\sigma_j^y P_{j+1} \\
+ \frac{1}{4g} P_{j-1}P_{j+1}, \rho \Bigg\} + \frac{1}{2g} P_{j-1}\sigma_j^y P_{j+1} \rho P_{j-1}\sigma_j^y P_{j+1} \Bigg],
\end{aligned}
\tag{A5}
$$

where $\{\cdot,\cdot\}$ is an anti-communication bracket and the last term in Eq. (A5) instead is the leading effective jump term. By ignoring the new quantum jump terms in Eq. (A5), we obtain an effective non-Hermitian Hamiltonian

$$
H_{\text{eff}} = \sum_i (P_{i-1}\sigma_i^x P_{j+1} - i\frac{\gamma}{8g^2} P_{i-1}\sigma_i^y P_{i+1} - i\frac{\gamma}{4g} P_{i-1}P_{i+1} + h\sigma_j^z).
\tag{A6}
$$

Clearly, non-Hermiticity gets suppressed when $g$ increases. Thus, the corresponding open system can show a significant weak-ergodicity breaking with a long-time revival of an initial state in Fig. 6 of the main text. In this sense, we conclude that the non-Hermitian Hamiltonian in Eq. (5) of the main text offers instructive insights for understanding the quantum many-body scarred dynamics in an open system based on the well-established correspondence.

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
