# Peer review of "Weak Ergodicity Breaking in Non-Hermitian Many-body Systems"

_SciPost Physics, doi:SciPost Phys. 15, 052 (2023)_

## Round 3 · Referee Report · Anonymous (Referee 3) · 2023-3-23

Report

The authors have addressed my concerns. I recommend publication.

---

## Round 3 · Referee Report · Anonymous (Referee 4) · 2023-5-15

Strengths

1. Interesting model to explore.

2. Highlights potentially interesting connection between ground state phase diagram and quantum many-body scars.

3. This version well-written with clear definitions.

Report

The authors have changed many parts of the manuscript in order to address the previous referee reports. The paper now contains two independent studies on a non-Hermitian extension of the PXP model, one on the ground state phase diagram, and one on properties of the higher spectrum such as level statistics and Quantum Many-Body Scars (QMBS). They also highlight some intriguing connections between these two properties, such as the enhancement of QMBS close to the exceptional point, which might encourage further studies of these phenomena. These changes address my previous concerns adequately, and I recommend the publication of the current version of the manuscript in SciPost.

---

## Round 3 · List of Changes

1. We have distinguished scarring/excited-state and ground-state properties in the revised manuscript. For instance, we have deleted the words “non-Hermitian QMBS” in Fig. 1, completely disconnecting the ground-state diagram from the properties of excited states.
2. We have replaced the statistics of energy level spacing at h=-0.3, γ=0.5 with that at h=0, γ=0.1 in Fig. 5a to better compare with Fig. 4a with h=0, γ=0.1.
3. We have improved the presentation in our manuscript to further clarify the definition of the “ground state” in section 3.1, “overlaps” in sections 3.2, 3.3 and the caption of Fig. 4, “bi-orthogonality” in section 2, "exceptional points" in section 1 and 3.1, different types of entanglement entropy in section 3.3.
4. In section 3.1, we have clarified the definitions of "critical exponents" and "central charge" and compared them with the Hermitian counterparts with the conclusion that the central charge is in very good agreement with an Ising universality class at the phase transition. As for "susceptibility", in section 3.1 of the revised manuscript, we have clarified more on its definition and highlighted that the negative divergence of the fidelity susceptibility in our model implies exceptional points, which are impossible to exist in any of the standard Hermitian systems.
5. In the section “introduction” of the revised manuscript, we have modified the sentence regarding the weak ergodicity breaking in non-Hermitian systems.
6. We have commented on the jump operators we constructed in relation to experimental realizations in Section 3.5, and emphasized experimentally realizable Lindblad operators in the section “Summary and Discussion”.
7. We have highlighted the research direction on the eigenmodes of the Liouvillian with purely imaginary eigenvalues from the perspective of the non-Hermitian QMBS.

---

## Editorial Decision

published